# Overexpression of BoLSU1 and BoLSU2 Confers Tolerance to Sulfur Deficiency in Arabidopsis by Manipulating Glucosinolate Metabolism

**DOI:** 10.3390/ijms241713520

**Published:** 2023-08-31

**Authors:** Shuang Yang, Zihuan Zhou, Tianqi Zhang, Qianqian Zhang, Rui Li, Jing Li

**Affiliations:** College of Life Sciences, Northeast Agricultural University, Harbin 150030, China; yangshuang@neau.edu.cn (S.Y.);

**Keywords:** *Brassica oleracea* var. *italica*, low sulfur responsive gene, sulfur metabolism, glucosinolates, glutathione

## Abstract

Sulfur is an essential element for plant growth, development and resistance to environmental stresses. Glucosinolates (GSLs), a group of sulfur rich secondary metabolites found in Brassicaceae plants, are known for their defensive properties against pathogens and herbivores. Due to their integration of a large proportion of total sulfur, their biosynthesis and degradation are closely linked to sulfur metabolism. It has been demonstrated that GSLs can be broken down to release sulfur and facilitate the production of other thio-metabolites when the plant is under stress. However, the regulation of this process is still not fully understood. In this study, we constructed two broccoli LSU (low sulfur responsive) gene overexpressing lines, *35S::BoLSU1* and *35S::BoLSU2*, to detect changes in GSL metabolism after sulfur deficiency treatment. The results showed that BoLSU1 and BoLSU2 inhibit the biosynthesis of aliphatic GSLs, while also promoting their degradation and increasing the content of glutathione (GSH), leading to the reallocation of sulfur from the GSL pool to other thio-metabolites such as GSH. Furthermore, this regulation of GSL metabolism mediated by BoLSU1 and BoLSU2 is found to be dependent on myrosinases BGLU28 and BGLU30. Our study provides insight into the physiological role of LSU proteins and their regulation of sulfur metabolism.

## 1. Introduction

Sulfur (S) is an essential macronutrient element for plant growth and development. It is a component of the amino acids cysteine and methionine, which are not only the building blocks of proteins but also the precursors of various sulfur-containing metabolites, including coenzyme A, vitamins and cofactors such as thiamine and biotin [1,2]. Thus, sulfur plays an important role in the metabolism of protein, sugar and fat. Furthermore, when plants are exposed to biotic or abiotic stress, sulfur-rich compounds such as H_2_S, glutathione (GSH), phytochelatins (PCs), metallothioneins (MTs), plant defensins and secondary metabolite glucosinolates (GSLs) are induced, which have been shown to be beneficial in responding to various environmental stresses [3,4,5]. When plants encounter sulfur deficiency, their growth and development are hindered and their resistance to stress also decreases [6].

GSLs are a class of sulfur-containing secondary metabolites common in Brassicacea species. Structurally, they are composed of an S-linked thioglucose unit, an O-sulfated (*Z*)-thiohydroximate group and a side chain derived from an amino acid [7]. GSLs can be further categorized into aliphatic, indolic and aromatic groups based on the structures of the amino acid side chains [7]. Since the GSL molecules contain 2–3 sulfur atoms, their biosynthesis and accumulation require a large amount of sulfur. In Brassicaceae plants, the sulfur integrated by GSLs makes up 30% or even more of the total sulfur [8]. Therefore, the biosynthesis and degradation of GSLs significantly affects the metabolism of S-containing compounds as well as the reallocation of S among these compounds.

It is generally believed that intact GSLs exhibit no direct bioactivity and need to be degraded by myrosinases into various physiologically active compounds [9], which are well-known to play a crucial role in the defensive response against biotic stresses such as herbivores and pathogens [10,11]. In addition to S, GSLs also contain nitrogen and glucose, so its biosynthesis consumes nutrient elements and energy [12]. Therefore, it is of general interest as to whether GSLs can be reused as reservoirs of N or S when plants are not under biotic stress. The first biochemical evidence that GSLs can be recycled as a S reservoir was provided by Sugiyama et al. [13]. They proved that two myrosinases, BGLU28 and BGLU30, are induced by sulfur deficiency and initiate the hydrolysis of aliphatic GSLs, with the released S being reintegrated into cysteine [13,14]. Additionally, Li et al. reported that GSLs can also be degraded as a source of S under cadmium stress, promoting the production of S-containing detoxifying metabolites such as GSH, PCs and MTs and improving cadmium tolerance [15]. Therefore, the involvement of GSLs as a sulfur source in the regulation of S metabolism may be a universal mechanism in Brassicacea plants. However, the exact regulators and mechanisms involved remain unknown. 

Response to low sulfur (*LSU*) genes are widely distributed in the plant kingdom and are known for their strong induction in response to sulfur deficiency [16]. Arabidopsis possesses four *LSU* genes: *LSU1*, *LSU2*, *LSU3* and *LSU4.* In addition to being induced by sulfur deficiency, *LSU1* and *LUS2* are also up-regulated by various stresses, including salt stress, iron deficiency, copper excess or basic pH, indicating their role in adapting to adverse environments [17,18,19]. LSU proteins are considered to be involved in multiple protein-protein interactions and have been predicted as hubs in the protein interaction network in Arabidopsis; they are also predicted to play an important role in coordinating plant resistance to stress [18,20]. A few studies on the physiological function of LSU proteins in Arabidopsis have been described. LSUs stimulate the production of H_2_O_2_ and potentially other ROS and induces stomatal closure to improve disease resistance [18]. It was found that *LSU2* is induced by *Pseudomonas syringae* infection and positively regulates the tolerance to this pathogen [21]. At present, the exact molecular function of LSUs remains uncertain, and studies on its physiological function are also quite limited. 

In this study, we found that two Broccoli *LSU* genes, *BoLSU1* and *BoLSU2*, are induced by sulfur deficiency. Overexpressing *BoLSU1* and *BoLSU2* in Arabidopsis increases tolerance to sulfur deficiency in a BGLU28 and BGLU30 dependent manner. Under sulfur deficiency, BoLSU1 and BoLSU2 inhibit the biosynthesis of and promote the degradation of aliphatic GSLs, which limits sulfur consumption and promotes sulfur release. The regulation of BoLSU1 and BoLSU2 on GSL metabolism facilitates S to flow from GSLs to another thio-metabolite GSH and thereby alleviating sulfur deficiency stress. Our study contributes to the understanding of GSLs recycling and its regulation under abiotic stress in Brassicacea plants.

## 2. Results

### 2.1. The Expression of BoLSU1 and BoLSU2 Is Induced by Sulfur Deficiency

According to the sequences of the unigene *LSU1* and *LSU2* in our previous analysis of the broccoli transcriptome [22], coding sequences (CDS) of the two broccoli *LSU* genes, BolC8t50074H and BolC7t44833H, were amplified (Appendix A). The obtained genes have 83.1% and 87.7% nucleotide homology with *AtLSU1* and *AtLSU2*, respectively, and were thus designated *BoLSU1* and *BoLSU2*. Previous studies revealed that *AtLSU1* and *AtLSU2* are strongly induced in sulfur-deficient conditions [16]. However, the expression of the *LSU* genes is rarely reported in other plant species. Using Arabidopsis transformed with *ProBoLSU1::GUS* and *ProBoLSU2::GUS* constructs, in which *GUS* was expressed driven by the *BoLSU1* and *BoLSU2* promotors, we examined the expression of *BoLSU1* and *BoLSU2*. Under normal conditions, the expression of *BoLSU1* and *BoLSU2* was relatively weak in the vascular bundle. However, under sulfur deficiency conditions, the expression of *BoLSU1* and *BoLSU2* was significantly increased (Figure 1A). This was further confirmed by qRT-PCR analysis in broccoli, where *BoLSU1* and *BoLSU2* expression was induced in response to sulfur deficiency, particularly *BoLSU2*, which experienced a 10-fold increase (Figure 1B).

### 2.2. Overexpression of BoLSU1 and BoLSU2 Enhanced Tolerance to Sulfur Deficiency

To explore the roles of BoLSU1 and BoLSU2 in sulfur deprivation, we constructed transgenic plants overexpressing either *BoLSU1* and *BoLSU2* (*35S::BoLSU1* and *35S::BoLSU2*) (Figure 2A). Wild-type and transgenic seedlings were then grown simultaneously on a normal and sulfur deficient media, respectively. In normal conditions, no significant differences were observed in *35S::BoLSU1* and *35S::BoLSU2* compared to wild-type. However, under sulfur deficiency, the wild-type plants displayed a decrease in fresh weight and root length while *35S::BoLSU1* and *35S::BoLSU2* were significantly less affected (Figure 2B,C).

### 2.3. Overexpression of BoLSU1 and BoLSU2 Alleviated Oxidative Stress Induced by Sulfur Deficiency

To investigate the role of BoLSU1 and BoLSU2 in photosynthesis and reactive oxygen species production and scavenging, we measured the chlorophyll content and antioxidant enzyme activity in wild-type and transgenic plants. As shown in Figure 3A, sulfur deficiency led to a decrease in total chlorophyll content in all the three genotypes. However, the total chlorophyll content in *35S::BoLSU1* and *35S::BoLSU2* was higher than that of the wild-type.

Malondialdehyde (MDA) content is an indicator of membrane lipid peroxidation. Figure 3B showed that sulfur deficiency resulted in an increase in MDA content; however, this increase was relatively small in *35S::BoLSU1* and *35S::BoLSU2*, indicating that the overexpression of *BoLSU1* and *BoLSU2* had a protective effect the cell membrane.

As an important reactive oxygen species, H_2_O_2_ level in sulfur deficiency in wild-type and transgenic plants was in agreement with MDA, indicating that overexpression of *BoLSU1* and *BoLSU2* enhances the ability to scavenge H_2_O_2_ (Figure 3C).

In addition, we also detected the activity of antioxidant enzymes, including Superoxide Dismutase (SOD), Peroxidase (POD), and Catalase (CAT) (Figure 3D–F). A significant increase in antioxidant enzyme activity under sulfur deficiency was observed. Compared to wild-type, the increase of antioxidant enzyme activity in *35S::BoLSU1* and *35S::BoLSU2* were greater. These results indicate that overexpression of *BoLSU1* and *BoLSU2* can promote antioxidant enzyme activity to alleviate oxidative stress caused by sulfur deficiency.

### 2.4. BoLSU1 and BoLSU2 Manipulated Glucosinolate Metabolism under Sulfur Deficiency

As sulfur rich secondary metabolites, glucosinolates (GSLs) have previously been reported to be exploited as a sulfur reservoir under sulfur deficiency [13]. To investigate whether BoLSU1 and BoLSU2 alleviate damage caused by sulfur deficiency through manipulating GSL metabolism, we detected the content of GSLs and the expression of genes in GSL metabolism pathway in *35S::BoLSU1* and *35S::BoLSU2* plants grown in hydroponics with or without sulfur. Under normal sulfur supply, the levels of all GSLs with different structures were similar between *35S::BoLSU1*, *35S::BoLSU2* and wild-type (Figure 4). However, under sulfur deficiency, both aliphatic and indolic GSLs decreased significantly. For aliphatic GSLs (including the ones with different side chain structure and chain length), the decline in *35S::BoLSU1* and *35S::BoLSU2* was significantly higher than that in wild-type. For indolic GSLs, the decrease in *35S::BoLSU1* and *35S::BoLSU2* was comparable to that in wild type. These results indicated that under sulfur deficiency, overexpression of *BoLSU1* and *BoLSU2* positively regulates the accumulation of aliphatic GSLs but does not affect the accumulation of indolic GSLs.

The decline of GSLs content induced by sulfur deficiency may be caused by either a decrease in biosynthesis or an increase in degradation. To investigate the effect of sulfur deficiency on GSL metabolism, we analyzed the expression levels of genes involved in the GSLs metabolic pathway under normal and sulfur deficiency conditions (Figure 5 and Figure 6). Under normal conditions, the transcript levels of genes involved in the aliphatic GSLs biosynthesis, including *MYB28*, *MYB29*, *MYB76*, *BCAT4*, *MAM1*, *CYP79F1*, *CYP79F2*, *CYP83A1*, and *FMO_GSOX1_*, were not altered in *35S::BoLSU1* and *35S::BoLSU2* compared to wild-type. However, in the absence of sulfur, the expression of these genes were significantly inhibited and were more strongly inhibited in *35S::BoLSU1* and *35S::BoLSU2*.

Myrosinases TGG1, BGLU28 and BGLU30 have been reported to catalyze the degradation of aliphatic GSLs. As shown in Figure 5, sulfur deficiency caused a notable increase in the expression of *BGLU28* and *BGLU30* in both wild-type and *BoLSU* overexpressing plants. In the wild-type, the expression levels of *BGLU28* and *BGLU30* increased 5 and 6 times, respectively. Whereas in overexpression lines, the expression levels of *BGLU28* and *BGLU30* increased to 18 and 24 times and 46 and 58 times, respectively, in *35S::BoLSU1* and *35S::BoLSU2*. Interestingly, the expression of *TGG1* was not induced by sulfur deficiency, and the expression level of *TGG1* in *35S::BoLSU1* and *35S::BoLSU2* were comparable to the wild-type under both normal or sulfur deficiency conditions.

The above results indicated that sulfur deficiency inhibited the biosynthesis and simultaneously promoted the degradation of aliphatic GSLs. Overexpression of *BoLSU1* and *BoLSU2* further enhanced the inhibition of aliphatic GSL biosynthesis and the promotion of aliphatic GSL degradation, which suggested that BoLSU1 and BoLSU2 prevent the flow of S to GSLs pool and stimulated the release of S from GSLs to cope with the sulfur deficiency.

As described above (Figure 4), the indolic GSLs content decreased under sulfur deficiency. However, the content of these compounds in *35S::BoLSU1* and *35S::BoLSU2* showed no significant difference from that in wild-type regardless of S supply, which indicated that BoLSU1 and BoLSU2 might not affect the metabolism of indolic GSLs. To confirm this hypothesis, we detected the expression of the key regulator and enzyme genes involved in indolic GSLs biosynthesis pathway, including *MYB51*, *MYB34*, *CYP79B2* and *CYP79B3*. As shown in Figure 6, the expression of the key genes in indolic GSLs biosynthesis pathway was inhibited by sulfur deficiency in both wild-type and overexpression lines. Interestingly, the expression of these genes was less inhibited in *35S::BoLSU1* and *35S::BoLSU2* than in wild-type, suggesting that under sulfur deficiency, BoLSU1 and BoLSU2 positively regulated the biosynthesis of indolic GSLs, which was inconsistent with our hypothesis.

Figure 6 revealed that the expression of *PEN2* and *PYK10*, which encode myrosinases that specifically catalyze the degradation of indolic GSLs, was not induced by sulfur deficiency and was also not altered in *35S::BoLSU1* and *35S::BoLSU2* under both normal and sulfur deficiency conditions. This implies that these two myrosinases were not regulated by BoLSU1 and BoLSU2 and did not participate in the response to sulfur deficiency.

Ryosuke Sugiyama et al. reported that BGLU28 and BGLU30 can not only catalyze the degradation of aliphatic GSLs, but also the degradation of indolic GSLs [13]. As mentioned above, in response to sulfur deficiency, BoLSU1 and BoLSU2 promoted the expression of genes involved in indolic GSLs biosynthesis and myrosinase genes *BGLU28* and *BGLU30*, indicating that BoLSU1 and BoLSU2 can simultaneously promote the production and degradation of indolic GSLs. This explains why the indolic GSLs content in *35S::BoLSU1* and *35S::BoLSU2* remained unchanged.

### 2.5. BoLSU1 and BoLSU2 Mediated Sulfur Deficiency Tolerance Depends on BGLU28 and BGLU30

Sulfur deficiency led to the induction of *BGLU28* and *BGLU30*, which was more strongly induced in *35S::BoLSU1* and *35S::BoLSU2* than in the wild-type, implying that BGLU28 and BGLU30 may be essential for BoLSU1- and BoLSU2-mediated sulfur deficiency tolerance. To test this hypothesis, we generated transgenic Arabidopsis plants overexpressing *BoLSU1* and *BoLSU2* in the *bglu28bglu30* background.

Under normal conditions, no considerable difference was observed between wild-type, *bglu28bglu30, 35S::BoLSU1, 35S::BoLSU2, 35S::BoLSU1/bglu28bglu30* and *35S::BoLSU2/bglu28bglu30* (Figure 7A). However, when S supply is lacking, all genotypes showed a decrease in fresh weight and root length, with the *bglu28bglu30* displaying a greater decrease compared to wild-type (Figure 7B). This suggested that BGLU28 and BGLU30 are essential in resisting sulfur deficiency stress. In addition, overexpressing *BoLSU1* and *BoLSU2* in the *bglu28bglu30* background did not improve sulfur deficiency tolerance as much as in wild-type background (Figure 7A,B), suggesting that BoLSU1- and BLSU2-mediated sulfur deficiency tolerance depends on BGLU28 and BGLU30.

### 2.6. Overexpression of BoLSU1 and BoLSU2 Increased the Content of Sulfur Metabolite GSH

GSH is an important sulfur-containing metabolite composed of three amino acids: glycine, cysteine and glutamic acid. The thiol group of cysteine residue in GSH renders it an effective antioxidant [23]. During the process of scavenging ROS, GSH is oxidized to form oxidized glutathione (GSSG) and GSH can be regenerated from GSSG. A recent study has demonstrated that GSLs are exploited as a sulfur source by the reallocation of sulfur atoms to primary metabolites such as GSH and Cys.

To investigate whether BoLSU1 and BoLSU2 mediated degradation of GSLs promotes S flow to GSH, we examined the content of GSH and GSSG in *35S::BoLSU1* and *35S::BoLSU2*. Under sulfur deficiency conditions, the content of GSH, as well as the GSH pool (total content of GSH and GSSG), significantly decreased, and a lower decrease was detected in *35S::BoLSU1* and *35S::BoLSU2* compared to wild-type plants (Figure 8A,C). In contrast, the content of GSSG were more decreased in *35S::BoLSU1* and *35S::BoLSU2* than in wild-type (Figure 8B). Combined with the above results, it can be inferred that overexpression of *BoLSU1* and *BoLSU2* promotes the degradation of GSLs and the release of sulfur, thus promoting the flow of sulfur from the GSLs pool to the GSH pool, and thus enhancing the antioxidant capacity and improving the tolerance to sulfur deficiency.

## 3. Discussion

In Brassicaceae plants, to a certain extent, increasing sulfur levels can promote the biosynthesis and accumulation of GSLs [8,24]. On the other hand, a recent study shows that when S supply is deficient, myrosinases BGLU28 and BGLU30 will be induced, causing GSLs to break down and the released S to be integrated into primary metabolite cysteine [13]. Furthermore, GSLs can also be degraded when exposed to heavy metal cadmium, promoting the production of sulfur-containing detoxifying substances such as GSH and PC, which alleviate the toxicity of cadmium [25]. Therefore, GSL’s role as a sulfur reservoir is important. However, the regulation of this process remains unclear.

In this study, we found that overexpression of *BoLSU1* and *BoLSU2* improves the tolerance of Arabidopsis to sulfur deficiency by manipulating metabolism of aliphatic GSLs. Combined with previous studies, a model of BoLSU1/2 regulating GSLs metabolism under sulfur deficiency was established. As illustrated in Figure 9, the biosynthesis of aliphatic GSLs requires a substantial amount of sulfur, since the precursor methionine, and the two sulfur donors GSH and PAPS (an intermediate product in the sulfur assimilation pathway), each contain sulfur atoms. Under sulfur deficiency, BoLSU1 and BoLSU2 inhibit the biosynthesis of aliphatic GSLs, which consequently reduce sulfur consumption. In addition, BoLSU1 and BoLSU2 improve tolerance to sulfur deficiency in a BGLU28 and BGLU30 dependent manner. Under sulfur deficiency, BoLSU1 and BoLSU2 induce the expression of myrosinase genes *BGLU28* and *BGLU30*, promoting the degradation of GSLs and thus accelerating the release of sulfur. In summary, overexpression of *BoLSU1* and *BoLSU2* prevents the influx of sulfur into the aliphatic GSLs pool and promotes the release of S from the GSLs pool, thus facilitating the integration of S into other thio-metabolites such as GSH (Figure 9).

The metabolism of GSLs is closely related to GSH. GSH directly participates in the biosynthesis of GSLs as a sulfur donor, and it also conjugates with ITC, the degraded product of GSL, for further hydrolysis [13,26]. Our study showed that overexpression of *BoLSU1* and *BoLSU2* decreased sulfur in the GSLs pool and at the same time increased sulfur in the GSH pool (the total of GSH and GSSG). Even though our data cannot prove whether the sulfur released by GSLs is directly incorporated into GSH, they demonstrate that the release of sulfur by GSLs will ensure more sulfur is available for GSH production. Due to its rich thiol groups, GSH acts as a universal scavenger of reactive oxygen species (ROS) and effectively mitigates oxidative stress-induced damage to plants. Therefore, the reallocation of sulfur between GSL and GSH manipulated by LSU proteins has a positive impact on plant resistance to environmental stress.

Our study sheds light on the role of LSUs in modulating sulfur metabolism, particularly glucosinolate metabolism. Nevertheless, there are still several important questions that remain unanswered. LSUs are believed to function through interactions with other proteins, so it is of great interest to know which proteins LSUs interact with to regulate glucosinolate metabolism. It is also worth exploring how the sulfur release from glucosinolates mediated by LSUs influences other thio-metabolites besides GSH.

*LSU* genes are widely conserved throughout angiosperms and they are activated in response to sulfur deficiency in various species [27], indicating that they have an important role in regulating sulfur metabolism across different species. Studies have demonstrated that in broccoli, similar to in Arabidopsis, sulfur supply has a great impact on GSLs content [28,29]. The proper application of sulfur fertilizer is beneficial to increase GSLs, while inadequate supply of sulfur not only restricts GSLs biosynthesis but also lead to GSLs degradation [30,31]. It can be seen that the regulation of GSLs and sulfur metabolism in Broccoli and Arabidopsis is quite conserved. Our study showed that overexpression of *BoLSU1* and *BoLSU2* improved tolerance to sulfur deficiency in Arabidopsis through manipulating GSL metabolism, which further demonstrates the conservation of GSL metabolism regulation in Arabidopsis and broccoli. Therefore, our study provides a valuable reference to gain a better understanding of the accumulation and regulation of GSLs in Arabidopsis as well as in broccoli.

## 4. Materials and Methods

### 4.1. Plant Materials and Culture Conditions

Seeds of broccoli (*Brassica oleracea* var. *italica*) cultivar ‘Youxiu’ (SAKATA, Yokohama, Japan) were used for gene cloning. The Arabidopsis used as wild-type (WT) in this study is Columbia-0 (Col-0). We obtained *bglu28* (CS25090) and *bglu30* (CS879652) from the Arabidopsis Biological Resource Center (Columbus, OH, USA). The *bglu28bglu30* double mutant line was obtained using Arabidopsis hybridization. All plants were cultivated under standard conditions with a 16-h-light/8-h-dark photoperiod at 23 °C and 60% relative humidity.

### 4.2. Sulfur Deficiency Treatment

Arabidopsis seeds were surface sterilized with a 70% ethanol solution for 30 s then subsequently sterilized with seed disinfectant (2% PPM, 0.1% TritonX-100) for 4–8 h. Replace the seed disinfectant and vernalize the seeds at 4 °C for 3 d. Seeds were cultured in 1/2MS medium (with 3% sucrose, pH = 5.8) [32] for 7 d and transferred to 12-well cell culture plates for sulfur deficiency treatment. Seedlings were treated with normal sulfur (+S, 1500 μM SO_4_^2−^) and sulfur deficiency (−S, 0 μM SO_4_^2−^) Hoagland culture solution [33] hydroponically, respectively.

### 4.3. Molecular Cloning and Plant Genetic Transformation

The promoters of *BoLSU1* and *BoLSU2* were amplified from the genomic DNA of broccoli seedlings using the primers ProBoLSU1-F, ProBoLSU1-R, ProBoLSU2-F and ProBoLSU2-R (primer sequences were listed in Appendix A). The PCR products were first cloned into the pMD^TM^ 18-T vector (TaKaRa, Beijing, China) and then cloned into the expression vector pCAMBIA3300NLS-GUS using the USER method as described previously [34]. *ProBoLSU1::GUS* and *ProBoLSU2::GUS* transgenic Arabidopsis were generated through *Agrobacterium tumefaciens* (GV3101)-mediated transformation using the floral dip method [35].

The trizol method was used to extract total RNA from broccoli seedlings treated with −S for 2 d. The total RNA extracted was reverse transcribed with a ReverTra Ace qPCR RT Master Mix with gDNA Remover Kit (TOYOBO, Shanghai, China) to obtain cDNA. The coding sequence of *BoLSU1 and BoLSU2* were amplified from the obtained cDNA using the primers *BoLSU1-F*, *BoLSU1-R*, *BoLSU2-F* and *BoLSU2-R* (primer sequences are listed in Appendix A). The PCR products were first cloned into the pMD^TM^ 18-T vector (TaKaRa, Beijing, China) and then cloned into the expression vector pCAMBIA330035Su using the USER method as described previously [34]. *35S::BoLSU1* and *35S::BoLSU2* transgenic Arabidopsis were generated through *Agrobacterium tumefaciens* (GV3101)-mediated transformation using the floral dip method [35]. The T3 homologous transgenic plants were utilized in the subsequent analyses.

### 4.4. GUS Detection

The histochemical GUS staining was performed as previously described [36]. Three lines were used for each sample, and at least 10 plants were observed for each line. The samples were observed under a stereomicroscope and subsequently photographed. The photographs show the most representative results.

### 4.5. Glucosinolate Extraction and Analysis

*35S::BoLSU1*, *35S::BoLSU2* and wild-type Arabidopsis seedlings were treated with sulfur deficiency for 7 d. Then, 150 mg seedlings were harvested for glucosinolate purification with reference to previous descriptions [37]. GSLs were extracted with 5 mL 80% pre-cooled methanol and the extraction passed through DEAE sephadex columns followed by sulphatase (Sigma, Shanghai, China) treatment. Sinigrin (Sigma, Shanghai, China) was used as an external standard. Extract solutions (5 μL) were subjected into Ultra-high-performance Liquid Chromatography (UPLC) (Agilent, 1290 infinity II) and GSLs were separated on ACQUITY UPLC^®^ HSS T3 (2.1 × 50 mm, 1.8 μm; Waters). GSL concentrations were normalized to fresh weight.

### 4.6. Quantitative Real-Time PCR Analyses

Total RNA of broccoli seedlings treated with normal sulfur and sulfur deficiency for 2 d was isolated using an Ultrapure RNA Kit (Cwbio, Suzhou, China). The synthesis of cDNA was performed using a ReverTra Ace qPCR RT Master Mix with gDNA Remover Kit (TOYOBO, Shanghai, China). Quantitative real-time PCR (qRT-PCR) was performed using a 2 × SYBR Green qPCR Mix (SparkJade, Dongying, China) on a QuantStudio 3 Real-Time PCR System (Thermo Fisher, Waltham, MA, USA). The *BoACTIN2* gene in broccoli was used as an internal control. Detection of *BoLSU1* and *BoLSu2* gene expression was performed with specific primers (primer sequences are listed in Appendix A).

Total RNA of Arabidopsis seedlings treated with normal sulfur and sulfur deficiency for 2 d was isolated using an Ultrapure RNA Kit (Cwbio, Suzhou, China). The synthesis of cDNA and quantitative real-time PCR (qRT-PCR) methods are the same as above. The *ACTIN2* gene in Arabidopsis was used as an internal control. Detection of *MYB28*, *MYB29*, *MYB76*, *BCAT4*, *MAM1*, *CYP79F1*, *CYP79F2*, *CYP83A1*, *FMO_GSOX1_*, *BGLU28*, *BGLU30*, *TGG1*, *MYB51*, *MYB34*, *CYP79B2*, *CYP79B3*, *PEN2*, and *PYK10* gene expression was done with specific primers (primer sequences were listed in Appendix A).

To ensure accuracy of data and statistical analysis, three independent biological replicates and three technical replicates were performed for each sample. The results were calculated using the 2^−ΔΔCT^ method [38].

### 4.7. Determination of GSH and GSSG Contents

*35S::BoLSU1*, *35S::BoLSU2* and wild-type Arabidopsis seedlings were treated with sulfur deficiency for 7 d. Total glutathione (GSH and GSSG) and oxidized glutathione (GSSG) were determined using an assay kit (Nanjing Jiancheng Bioengineering Institute, A061-2-1, Nanjing, China). The assay was based on the DTNB (5,5’-dithiobis-2-nitrobenzoic acid) recycling reaction to determine both total glutathione and GSSG contents. GSH reacted with DTNB to produce the yellow product 2-nitro-5-mercaptobenzoic acid, which had the maximum light absorption at 412 nm. The content (μg/g FW) of total glutathione and GSSG were calculated in accordance with instruction. GSH content was estimated as subtraction of GSSG from total glutathione.

### 4.8. Measurement of Chlorophyll Content

*35S::BoLSU1*, *35S::BoLSU2* and wild-type Arabidopsis seedlings were treated with sulfur deficiency for 7 d. 100 mg seedlings were frozen separately in liquid nitrogen and homogenized with liquid nitrogen. Chlorophyll content was determined using an assay kit (Nanjing Jiancheng Bioengineering Institute, A147-1-1, Nanjing, China), and chlorophyll was extracted with a mixed solution of ethanol and acetone (1:2). Measurements were made at 645 nm and 663 nm. Chlorophyll content was calculated in accordance with manufacturer’s instruction.

### 4.9. Antioxidant Enzyme Activity Assay

*35S::BoLSU1*, *35S::BoLSU2* and wild-type Arabidopsis seedlings were treated with sulfur deficiency for 7 d then separately homogenized by tissue crusher (40 Hz, 5 min). Peroxidase (POD) activity was determined using an assay kit (Nanjing Jiancheng Bioengineering Institute, A084-3-1, Nanjing, China). Measurements were made at 420 nm. Superoxide Dismutase (SOD) activity was determined using an assay kit (Nanjing Jiancheng Bioengineering Institute, A001-1-1, Nanjing, China) by Hydroxylamine method. Measurements were made at 550 nm. Malondialdehyde (MDA) activity was determined using an assay kit (Nanjing Jiancheng Bioengineering Institute, A003-3-1, Nanjing, China) by Colorimetric method. Measurements were made at 532 nm. Catalase (CAT) activity was determined using an assay kit (Nanjing Jiancheng Bioengineering Institute, A007-1-1, China). Measurements were made at 405 nm. H_2_O_2_ content were determined using an assay kit (Nanjing Jiancheng Bioengineering Institute, A064-1-1, China). Measurements were made at 405 nm. The calculation formula for each of the above enzyme’s activity is detailed in the instruction manual.

## Figures and Tables

**Figure 1 ijms-24-13520-f001:**
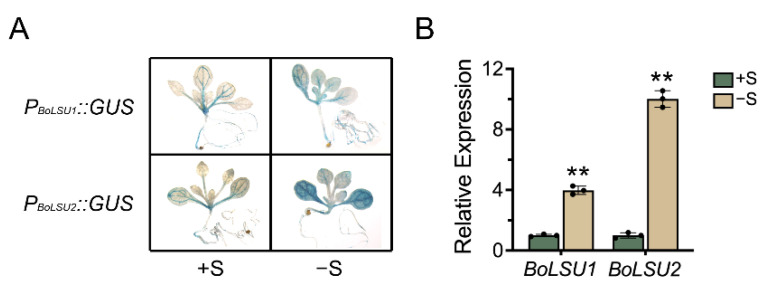
The expression of *BoLSU1* and *BoLSU2* under normal sulfur (+S) and sulfur deficiency (−S) conditions. (**A**) *BoLSU1* and *BoLSU2* promoter-directed *GUS* expression in Arabidopsis. (**B**) Relative expression of *BoLSU1* and *BoLSU2* under −S conditions in broccoli analyzed by qRT-PCR. Scatter plots indicate three biological replicates. Error bars indicate SE of triplicate replicates. Asterisks indicate significant differences between +S and −S, ** *p* < 0.01 by Student’s t test.

**Figure 2 ijms-24-13520-f002:**
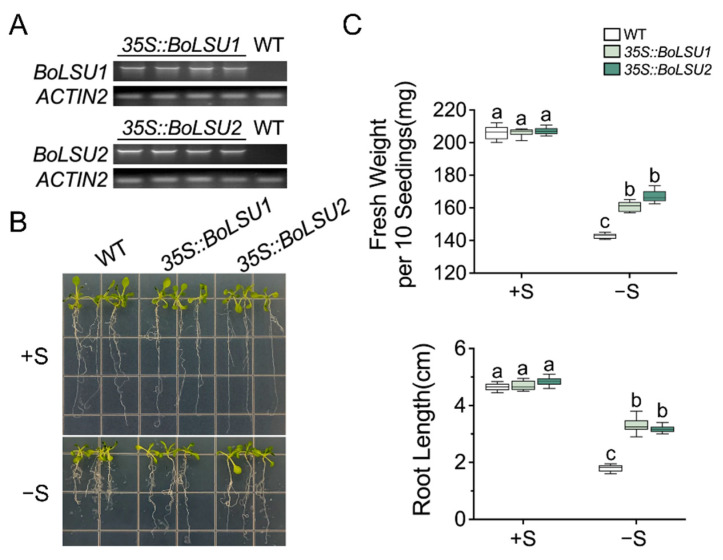
Analysis of sulfur deficiency tolerance in *35S::BoLSU1* and *35S::BoLSU2*. (**A**) Semiquantitative RT–PCR analysis of *BoLSU1* and *BoLSU2* expression. *ACTIN2* was used as an internal reference gene. (**B**) Growth of 14-day-old seedlings. (**C**) Fresh weight and root length of 14-day-old seedlings. Boxes indicate interquartile range, lines in box indicate median and whiskers indicate 1.5 times the extent of the interquartile range. Different letters indicate significant differences (Tukey’s post hoc test, *p* < 0.05).

**Figure 3 ijms-24-13520-f003:**
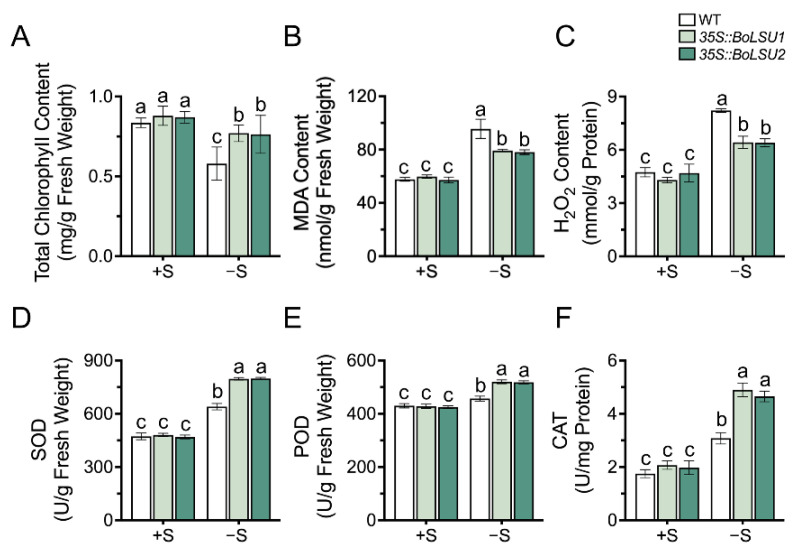
The effect of sulfur deficiency (−S) on total chlorophyll content (**A**) Malondialdehyde content (**B**) H_2_O_2_ content (**C**) activity of Superoxide Dismutase (**D**) Peroxidase (**E**) and Catalase (**F**) in WT, *35S::BoLSU1* and *35S::BoLSU2*. Different letters indicate significant differences (Tukey’s post hoc test, *p* < 0.05).

**Figure 4 ijms-24-13520-f004:**
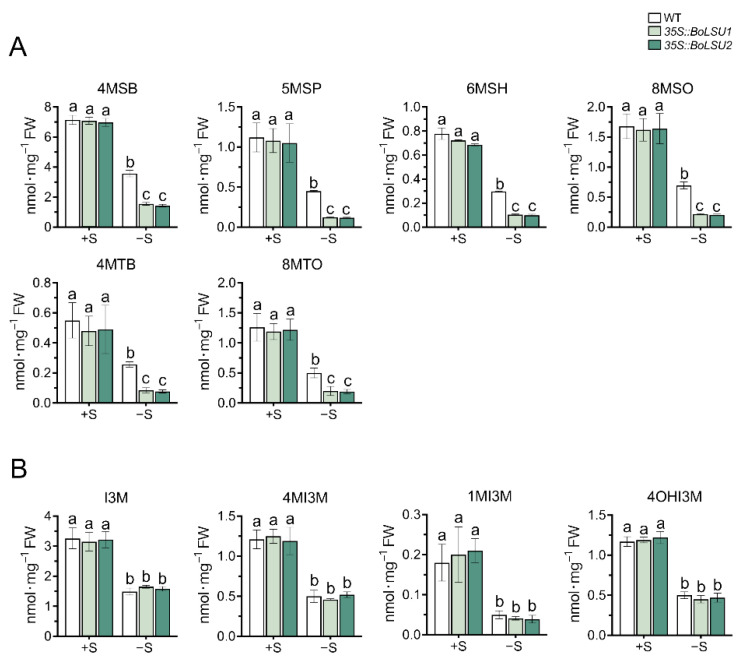
Effects of sulfur deficiency (−S) on glucosinolate content in WT, *35S::BoLSU1*, and *35S::BoLSU2*. (**A**) Effects of −S on aliphatic glucosinolate contents. 4MSB, 4-Methylsulfinyl-*n*-butyl glucosinolate; 5MSP, 5-Methylsulfinyl-*n*-pentyl glucosinolate; 6MSH, 6-Methylsulfinyl-*n*-hexyl glucosinolate; 8MSO, 8-Methylsulfinyl-*n*-octyl glucosinolate; 4MTB, 4-Methylthio-*n*-butyl glucosinolate; 8MTO, 8-Methylthio-*n*-octyl glucosinolate. (**B**) Effects of −S on indolic glucosinolate contents. I3M, Indol-3-ylmethyl glucosinolate; 4MI3M, 4-Methoxyindol-3-ylmethyl glucosinolate; 1MI3M, 1-Methoxyindol-3-ylmethyl glucosinolate; 4OHI3M, 4-Hydroxyindol-3-ylmethyl glucosinolate. Different letters indicate significant differences (Tukey’s post hoc test, *p* < 0.05).

**Figure 5 ijms-24-13520-f005:**
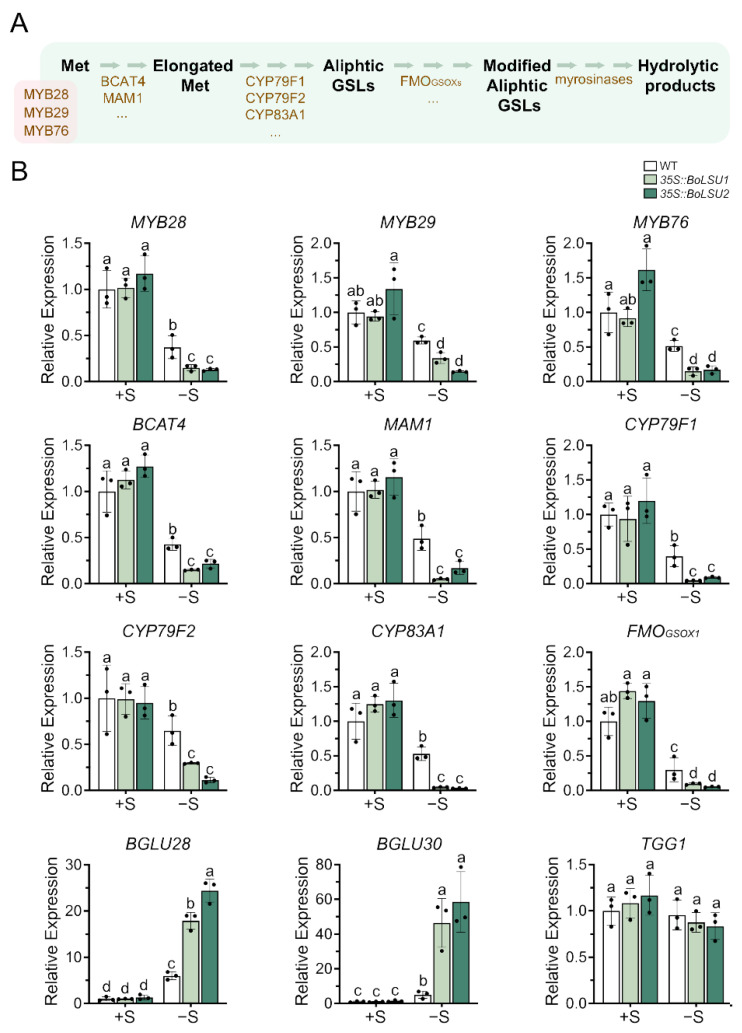
Expression of key genes in aliphatic glucosinolate metabolic pathway under normal sulfur (+S) and sulfur deficiency (−S) conditions. (**A**) Schematic diagram of aliphatic glucosinolate metabolic pathway. Met, methionine; GSLs, glucosinolates. Metabolites are depicted in black. Transcription factors (MYB28, MYB29 and MYB76) and enzymes are depicted in brown. (**B**) Relative expression level of genes of aliphatic glucosinolate metabolic pathway in WT, *35S::BoLSU1* and *35S::BoLSU2*. Scatter plots indicate three biological replicates. Different letters indicate significant differences (Tukey’s post hoc test, *p* < 0.05).

**Figure 6 ijms-24-13520-f006:**
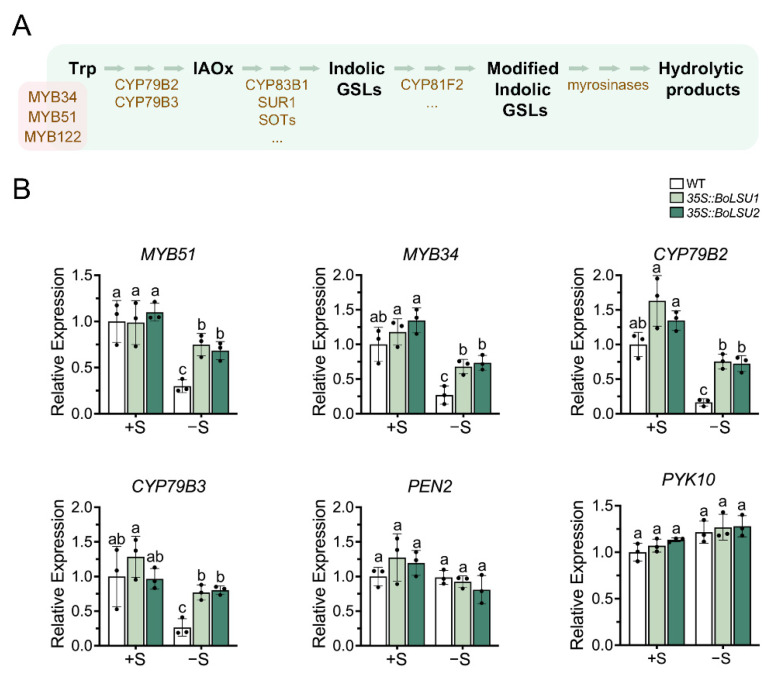
Expression of key genes in indolic glucosinolate metabolic pathway under normal sulfur (+S) and sulfur deficiency (−S) conditions. (**A**) Schematic diagram of indolic glucosinolate metabolic pathway. Trp, tryptophan; IAOx, Indole-3-acetaldoxime; GSLs, glucosinolates. Metabolites are depicted in black. Transcription factors (MYB34, MYB51 and MYB122) and enzymes are depicted in brown. (**B**) Relative expression level of genes of indolic glucosinolate metabolic pathway in WT, *35S::BoLSU1* and *35S::BoLSU2*. Scatter plots indicate three biological replicates. Different letters indicate significant differences (Tukey’s post hoc test, *p* < 0.05).

**Figure 7 ijms-24-13520-f007:**
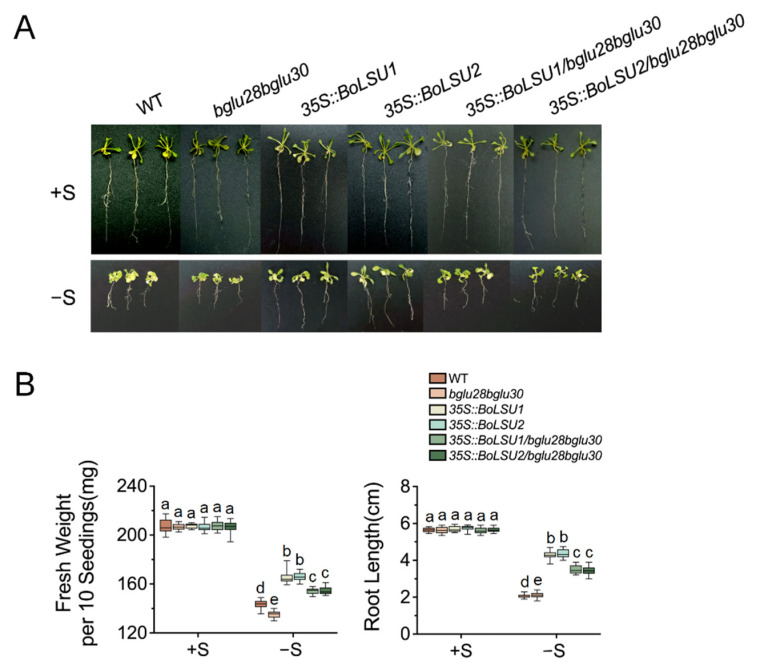
BGLU28 and BGLU30 are required for BoLSU1/2-mediated sulfur deficiency tolerance. (**A**) Phenotypes of WT, *bglu28bglu30*, *35S::BoLSU1*, *35S::BoLSU2*, *35S::BoLSU1*/*bglu28bglu30*, *35S::BoLSU2*/*bglu28bglu30* under normal sulfur (+S) and sulfur deficiency (−S) conditions. (**B**) Fresh weight and root length in different genotypes under +S and −S conditions. Boxes indicate interquartile range, lines in box indicate median, and whiskers indicate 1.5 times the extent of the interquartile range. Different letters indicate significant differences (Tukey’s post hoc test, *p* < 0.05).

**Figure 8 ijms-24-13520-f008:**
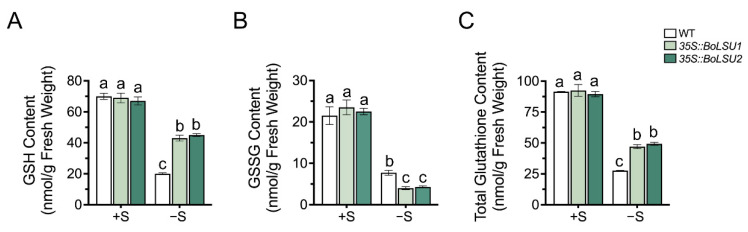
The effects of sulfur deficiency on GSH and GSH pool. (**A**) The effects of sulfur deficiency on GSH content (**B**) The effects of sulfur deficiency on GSSG content (**C**) The effects of sulfur deficiency on total GSH and GSSG content. Different letters indicate significant differences (Tukey’s post hoc test, *p* < 0.05).

**Figure 9 ijms-24-13520-f009:**
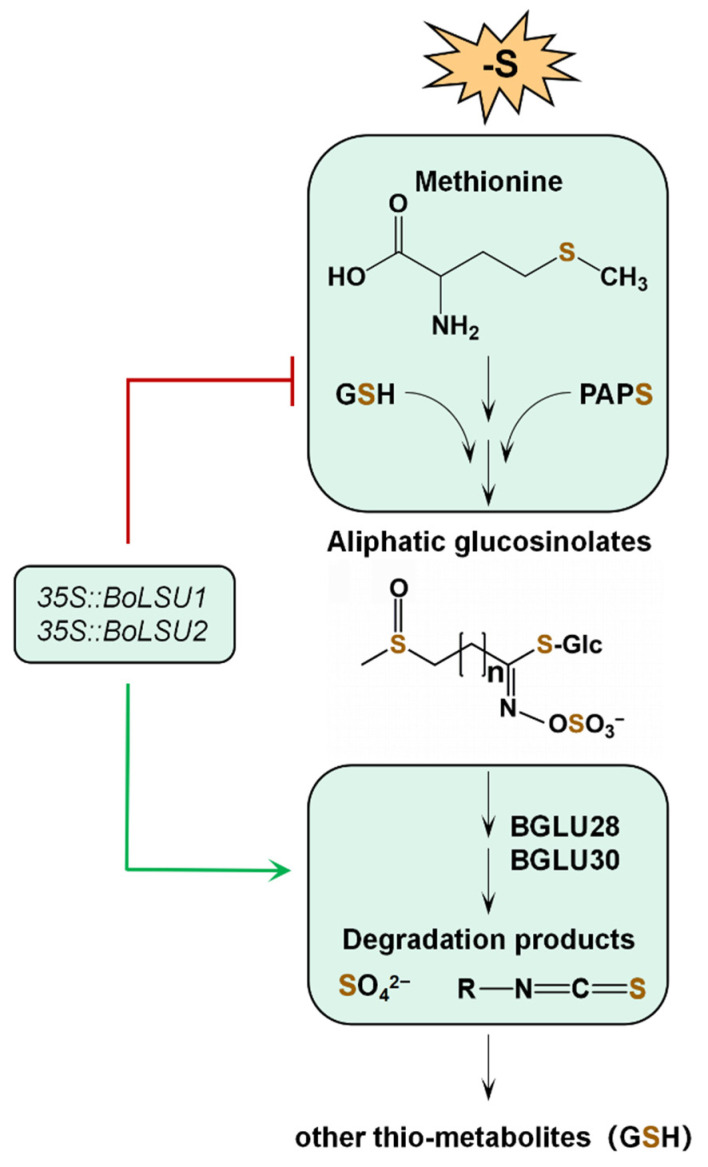
A model of BoLSU1 and BoLSU2 inhibiting aliphatic glucosinolates biosynthesis and promoting glucosinolate degradation to resist sulfur deficiency stress. GSH, glutathione; PAPS, 3′-phosphoadenosine-5′-phosphosulfate. Sulfur is shown in brown.

## Data Availability

Not applicable.

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
