# Peer review of "Overexpression of BoLSU1 and BoLSU2 Confers Tolerance to Sulfur Deficiency in Arabidopsis by Manipulating Glucosinolate Metabolism"

_ijms, 2023, doi:10.3390/ijms241713520_

Round 1
Reviewer 1 Report
This study describes the effects of the overexpression of LSU1/2 gene from broccoli in Arabidopsis thaliana. The authors claim that such genetic manipulation improves tolerance to sulfur deficiency by promoting degradation of glucosinolates resulting in increase of glutathione content. Despite the well-structured design of experiments and clear presentation, the depth of interpretation of findings and analysis of results could be further enriched. I have several concerns and suggestions, which should be addressed before the manuscript merits publication in JIMS.
1./ Are the changes observed specific for the broccoli LSUs overexpression? The authors just published a paper where they observed enhanced resistance to cadmium in AtLSUs overexpressants. I think it would be beneficial if they performed similar measurements (GSL genes expression, GSL content, GSH content) in sulfur deficiency conditions also for that lines just to test whether the effects they see are connected with specie-specific regulation.
2./ Are there only two genes encoding LSUs in broccoli? Arabidopsis has four of them.
3./ The article would benefit largely if the nucleotide sequences of the CDS of BoLSU1/2, as well as their promotors, would be added. Do promoters of BoLSU1/2 contain cis-acting sequences that allow for SLIM1 binding (necessary for –S dependent induction)? Do the genes contain introns (as may be assumed from the fact that the authors used cDNA instead of genomic DNA to amplify the CDSs).
4./ I am also surprised by the fact that Arabidopsis plants (WT) grown in sulfur deficiency (shown in Fig 2) do not accumulate anthocyanins. It is a common protective mechanism in plants grown under stress conditions.
5./ How would authors explain a stimulating effect of BoLSUs on antioxidant enzyme activities (Fig 3)? Do BoLSUs bind to the enzymes and increase their activity (as shown for FSD2 by Garcia-Molina et al, 2017). Or rather increase their amount by increasing transcription? I would like to see the transcript level of those enzymes.
6./ Also I find it a bit contradictory that BoLSUs overexpressing plants increase antioxidant enzyme activities, in the presence of the increased amount of glutathione (Fig 8), which is a known scavenger.
7./ Did the authors assay SDI1 expression? The effect they see on GSL level and their synthesis genes very much resembles the changes in SDI1_OX Arabidopsis (Aarabi et al, 2016). I think this is necessary control that needs to be added to fulfill the story.
8./ How do the authors explain the changes in so many gene expressions? How do BoLSUs affect transcription, do they interact with some transcription factors? Do they interact with BGLU28/30 proteins?
9./ Is the difference in root length shown in Fig 7B (-S) between the WT and bgl28/30 mutant statistically significant (the length looks the same looking at the graph).
10./ The authors should specify how did they get double bgl28/30 mutant in the ‘Plant material’ section.
11./ I do not understand the phenomenon of increasing the expression of indolic GSL synthesis genes together with the increase of their degradation in BoLSUs overexpressants in the conditions where sulfate is limiting. What would be the physiological meaning?
12./ A comprehensive discussion regarding the functional relevance of LSU genes in relation to the observed changes in sulfur metabolism during sulfur deficiency would offer a more holistic understanding of the author’s finding.
I also found a few language/typo mistakes so please correct them accordingly:
Line #113 remove the dot before ‘while’
Line #133 correct ‘response to’ to ‘level in’
Line #196 insert ‘to’ after ‘were comparable’
Line #294 correct ‘reducing’ to ‘reduce’
Line #354 give references for MS, as well as for Hoagland medium later on
Line #366 Instead of ‘use’ put ‘was used’
Line #382 put ‘were’ after seedlings
Line #397 the sentence makes no sense; rather write ‘detection of BoLSU1 and BoLSu2 gene expression was done with specific primers’
Line #400-404 the sentence makes no sense; see above
Line #409 put ‘were’ after seedlings
Line #419 put ‘were’ after seedlings
Line #426 put ‘were’ after seedlings
Line #420 put the weight of the seedlings used
Line #428-9 the sentence makes no sense
Line #429-37 put ‘activity’ after each enzyme name and change ‘were’ to ‘was’
Line #437 change the sentence to ‘The calculation formula for each of the above enzyme's activity is detailed in the instruction manual’
Reviewer 2 Report
Dear Editors,
Thank you so much for choosing me as a reviewer of the manuscript ijms-2552510 entitled: “Overexpression of BoLSU1 and BoLSU2 confers tolerance to sulfur deficiency in Arabidopsis by manipulating glucosinolate metabolism”. I hope that my comments will help Authors to improve their manuscript. Detailed remarks concerning the manuscript:
1. The clear purpose of the report and the scientific hypothesis with the clear answer to the question stated as a scientific hypothesis should be given
2. As it is suggested in the instructions for author the style of structured abstracts without headings should be used: (1) Background and highlight the purpose of the study; (2) Methods: briefly describe the main methods or treatments applied; (3) Results: summarize the article’s main findings; (4) Conclusions: indicate the main conclusions or interpretations. Please do needed changes
3. It is not recommended to use as key words the words or phrases used in the title of the manuscript. Please do needed changes.
4. There are manuscript citations included in the description of the results that sounds like discussion.
5. I suggest to divide ‘Discussion’ section into the subsections corresponded to the ‘Results’ subsection
6. conclusions are missed
7. The figures should be clear for the reader without necessary to refer to the text of the manuscript. Please give additional explanations where needed.
8. Bibliography should be prepared strictly to the guidelines for authors. Please do needed changes.
